# ATTRI-SSC-VAE: Multi-Attribute Regularized Sparse Coding VAEs for Interpretable Medical Image Representation

## Abstract

Explainable image representations are critical in medical imaging, where interpretability is essential for both clinical trust and decision-making. We introduce Attri-SSC-VAE, a novel framework that extends Structured Sparse Coding Variational Autoencoders (SSC-VAEs) with attribute regularization and multi-attribute mapping. Our approach leverages sparse coding to discretize image representations into a dictionary of latent components while preserving generative flexibility through a VAE encoder–decoder structure. To enhance interpretability, we impose attribute regularization on the coding coefficients, explicitly associating dictionary elements with meaningful clinical attributes. Furthermore, a multi-attribute mapping mechanism enables disentanglement across attributes, ensuring that variations in specific coding coefficients correspond to consistent and explainable changes in image features. This property allows for controlled image editing, where manipulating the coefficients associated with target attributes results in semantically aligned modifications in generated images. Experiments on medical imaging datasets demonstrate that Attri-SSC-VAE not only achieves competitive reconstruction and generation performance but also provides interpretable, attribute-aware representations that improve trustworthiness and practical utility in clinical applications.

## 1 Introduction

Latent image representation learning based on Variational Autoencoders (VAEs) has paved the foundation for many generative models by encoding data into meaningful low-dimensional latent vectors, typically drawn from a Gaussian prior. The Vector-Quantized VAE (VQ-VAE) (van den Oord et al., 2017) further advanced this line of work by replacing continuous latent variables with a discrete codebook of embeddings, enabling the model to preserve more information during image reconstruction and generation. While these frameworks provide powerful and compact representations, the roles of individual latent dimensions or codebook atoms are not directly interpretable. This makes it unclear how image attributes are encoded or how they influence reconstruction and generation. This limitation is particularly critical in medical image analysis, where interpretability is essential for doctors and patients to understand and trust model outputs. A central question is whether clinically meaningful attributes (e.g., anatomical or pathological features) can be explicitly embedded into hidden representations. Although such attributes are evident at the image level, current research has rarely established explicit alignment with latent variables or codebook atoms. This gap not only undermines explainability but also prevents direct control of image generation through specific clinical attributes.

Existing research has attempted to enhance the explainability through disentanglement methods (Higgins et al., 2017; Kim & Mnih, 2018; Chen et al., 2018; Locatello et al., 2018), which encourage factorization of the latent space so that each dimension encodes a distinct generative factor. However, purely unsupervised disentanglement is often fragile: results can vary depending on network architecture, hyperparameters, or random initialization, and some degree of supervision is typically required to obtain meaningful factors (Locatello et al., 2018). Moreover, since disentanglement is usually learned without explicit meanings corresponding to specific image attributes, post-

hoc analysis is usually needed to determine how specific attributes map to latent dimensions (Pati & Lerch, 2020).

Attribute-based methods (Hadjeres et al., 2017; Lample et al., 2017; Bouchacourt et al., 2017; Pati & Lerch, 2020) offer a more direct approach by explicitly associating latent dimensions with specific data attributes. Recent developments, such as Attri-VAE (Carter & Nielsen, 2017; Pati & Lerch, 2020), employ attribute-based regularization in the latent space of VAE to enhance interpretability, demonstrating encouraging results for controlled data generation and clinical attribute encoding. A major challenge still remains: VAEs map attributes to single latent dimensions, which often fails to capture the more complex and distributed nature of real-world attributes. In medical data, attributes such as left ventricle (LV) volume, myocardial volume, wall thickness, and so on, are continuous, physiologically correlated, and inherently multi-dimensional, meaning they cannot be fully captured by a single latent dimension, but rather a subspace.

To address these gaps, we propose Attri-SSC-VAE, a Structured Sparse Coding VAE with attribute regularization and multi-attribute mapping. The key idea is to associate each attribute not with a single latent dimension or isolated atom, but with groups of atoms in a sparse coding codebook. This design better reflects the distributed nature of real attributes, which often correspond to overlapping and correlated patterns rather than independent factors. By combining the discrete representational power of sparse coding VQ-VAEs with structured attribute regularization, Attri-SSC-VAE yields explainable medical image representations where clinical attributes are explicitly linked to interpretable latent structures. Furthermore, the framework supports attribute-driven image generation: editing the coding coefficients tied to specific attributes produces controlled and semantically aligned changes in generated images. Experiments on medical imaging datasets demonstrate that Attri-SSC-VAE not only achieves competitive reconstruction and generation performance with preserved fine-grained structures but also provides interpretable, attribute-aware representations, enabling trustworthy and clinically meaningful outcome.

Our contributions of the paper can be summarized:

- **Attibute-aware Fine-grained Representations:** Attri-SSC-VAE introduces attribute-regularized multi-atom coding, where each attribute is represented by a group of sparse codes. This design preserves fine structural details in reconstructed and generated images while yielding clinically meaningful and interpretable latent representations.
- **Controllable Generation and Editing:** The framework enables attribute-guided manipulation, where modifying coefficients associated with target attributes produces consistent and explainable changes in the generated images, supporting controllable image generation and editing.
- **Modeling Attribute Correlations:** Unlike one-to-one atom–attribute mappings, our approach allows dictionary atoms to overlap across attributes, enabling the model to naturally capture correlations between clinically related attributes, where shared atoms contribute jointly to multiple image attributes.

## 2 RELATED WORK

### 2.1 IMAGE REPRESENTATIONS

Image representation is critical in medical imaging, where fine structural details often carry diagnostic value. Variational Autoencoders (VAEs) (Kingma & Welling, 2014) compress images into smooth latent spaces but typically generate blurry reconstructions and struggle to disentangle factors of variation. Discrete latent models address these issues: Vector Quantized VAE (VQ-VAE) (van den Oord et al., 2017) encodes images into discrete codes via vector quantization, improving generation fidelity and controllability. Its extensions, such as VQ-VAE2 (Razavi et al., 2019) with hierarchical quantization and VQ-GAN (Esser et al., 2021) with adversarial training, further enhance detail preservation and realism, making them strong candidates for high-quality medical image reconstruction and synthesis.

Building on these advances, Sparse Coding VAE (SC-VAE) (Xiao et al., 2023) replaces single code assignments with sparse combinations of multiple dictionary atoms, boosting expressiveness and preserving local features. However, SC-VAE still treats atoms independently, limiting its ability to

capture relationships among them. Structured Sparse Coding VAE (SSC-VAE) (Wang et al., 2025) addresses this by explicitly modeling correlations between atoms through adaptive thresholds and attention, significantly improving fine-grained reconstruction and robust generation. These developments highlight the potential of discrete latent models for medical imaging, where both fidelity and interpretability are essential.

### 2.2 ATTRIBUTE EXPLAINABILITY

**Attribute-based Explanation in Neural Networks.** Post-hoc explanation methods such as saliency maps (Simonyan et al., 2013; Kapishnikov et al., 2019), Grad-CAM (Selvaraju et al., 2016), and concept activation tests (Goh et al., 2021) are widely used to audit neural networks, but they remain heuristic, brittle to perturbations, and do not enforce that human concepts are encoded in latent space. Quantitative Concept Activation Vectors (TCAV) (Kim et al., 2017) and Automatic Concept Discovery (ACE) (Ghorbani et al., 2019) provide concept-level analyses but operate after training and require concept exemplars or clustering heuristics, limiting principled interventions. Concept Bottleneck Models (CBMs) (Koh et al., 2020) and concept-whitening approaches (Chen et al., 2020) instead elevate concepts to first-class variables, enabling direct interventions and causal analyses; however, they typically demand strong supervision and assume near one-to-one alignment between concepts and latent units, which rarely holds in complex images.

**Attribute-Centric Generative Models.** Attribute-centric generative models explicitly inject attribute supervision into latent representations to enable controllable image generation. Early examples include Fader Networks (Lample et al., 2017) and AttGAN (He et al., 2017), which condition image generation on binary attributes, allowing targeted manipulation but often relying on simple one-attribute-per-dimension encoding. Attribute-regularized VAEs, such as Attri-VAE (Pati & Lerch, 2020; Cetin et al., 2022), enforce latent alignment with human-interpretable attributes, supporting concept-conditioned generation and attribute-guided editing. However, these models typically assume strong labels and one-to-one correspondence between latent dimensions and attributes, limiting their ability to represent multi-faceted, overlapping, or correlated attributes that may require multiple latent units to fully encode.

## 3 METHODOLOGY

Our framework provides an interpretable, attribute-aware representation of medical images by linking structured sparse codes with clinical attributes as shown in Figure 1. A 3D medical image is first encoded into feature maps and decomposed into sparse codes through a learned dictionary of latent atoms by SSC-VAE backbone. Atom activations are aggregated into usage vectors and mapped to clinical attributes via a sparsity-regularized mapper $W$, ensuring compact one-to-few attribute–atom associations. Attribute alignment further enforces consistency with ground-truth labels. The decoder then reconstructs the input for high-fidelity image recovery, while controlled modulation of attribute-specific atoms enables clinically meaningful, attribute-driven image generation and editing.

### 3.1 SSC-VAE

Given a 3D input $X \in \mathbb{R}^{C \times D \times H \times W}$, the encoder produces feature maps $E \in \mathbb{R}^{C' \times d \times h \times w}$. We learn a dictionary $D \in \mathbb{R}^{C' \times K}$ and sparse codes $Z \in \mathbb{R}^{K \times d \times h \times w}$, where each spatial location is represented as a sparse combination of dictionary atoms. The overall training objective combines reconstruction and latent regularization losses:

$$L_{\text{SSC}} = \mathcal{L}_{\text{recon}} + \mathcal{L}_{\text{latent}} \tag{1}$$

with

$$L_{\text{recon}} = \|G(E(X)) - X\|_2^2 \tag{2}$$

$$L_{\text{latent}} = \|E(X) - DZ\|_2^2 + \sum_i \alpha_{k,i} \|Z_{k,i}\|_1 , \tag{3}$$

where $\alpha \in \mathbb{R}^{K \times d \times h \times w}$ is a learnable threshold map that enforces location-aware sparsity, ensuring compact codes while preserving high-fidelity reconstructions. To capture correlations across

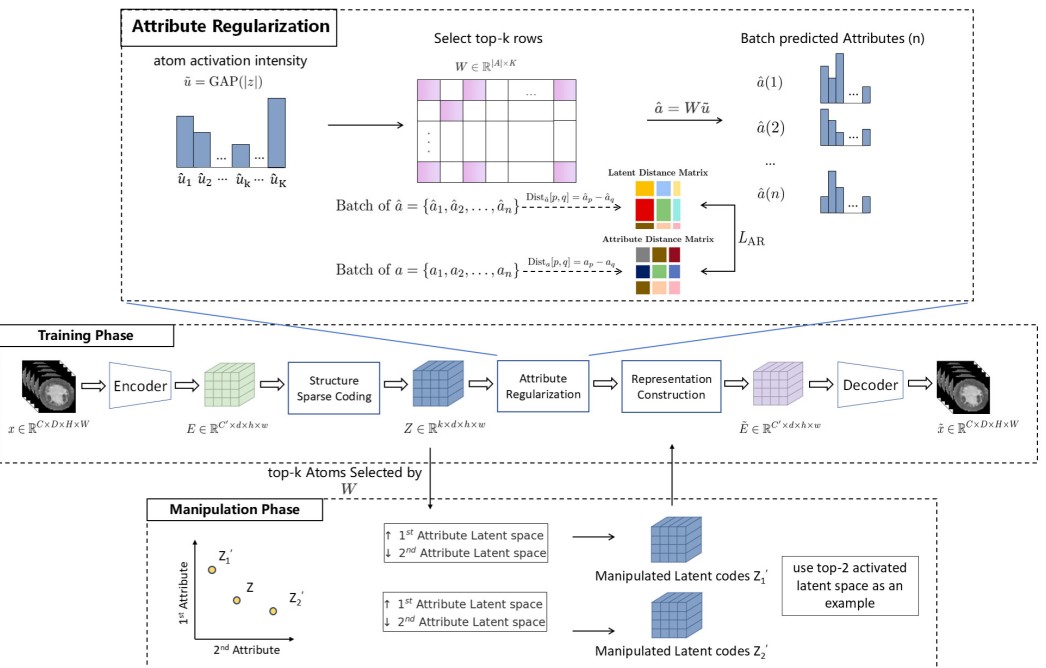

Figure 1: **Architecture of Attri-SSC-VAE.** The model links clinical attributes to the image latent representation via a sparse mapping $W$ between structured sparse codes $Z$ and the samples' attribute values in the "Attribute Regularization" module. Attribute usage is first summarized by applying global average pooling (GAP) over spatial dimensions on $Z$ to obtain atom activation intensities $\tilde{u}$ and then mapped to clinical attributes via a sparsity-regularized mapper $W$. AR loss is computed to align further the consistency of the reconstructed attribute values $\hat{a}$ with ground-truth labels $a$. The whole framework enables both high-fidelity reconstruction and controllable synthesis by manipulating the activations of semantically meaningful latent atoms.

dictionary atoms and spatial regions, $\alpha$ is refined via channel- and spatial-attention mechanisms:

$$W_c = \sigma(\text{MLP}(\text{AvgPool}(\alpha))) + \text{MLP}(\text{MaxPool}(\alpha)), \tag{4}$$

$$W_s = \sigma\left(f\left([\text{AvgPool}(\alpha); \text{MaxPool}(\alpha)]\right)\right), \tag{5}$$

$$\alpha^+ = W_s \odot (W_c \odot \alpha). \tag{6}$$

Detailed explanations of the above transformation functions of $W_c$ and $W_s$ to impose the correlations both in channels and image space can be found in Wang et al. (2025). The refined thresholds $\alpha^+$ are fed into an unfolded Learned Iterative Soft Thresholding Algorithm (LISTA) module Wang et al. (2025) to compute sparse codes $Z$ during the forward pass. Training updates the dictionary $D$ along with the encoder $E$ and decoder $G$ weights in the back propagation processing, enabling end-to-end learning of structured sparse representations.

## 3.2 MULTI-ATTRIBUTE MAPPING AND REGULARIZATION

To explicitly relate latent dictionary atoms to dataset attributes, we introduce a linear mapper $W \in \mathbb{R}^{|A| \times K}$, where $A$ denotes the set of supervised attributes and $|A|$ is its cardinality. The mapper projects atom usage patterns into the attribute space. For each sample, we first summarize its atom activations into a non-negative usage vector $u \in \mathbb{R}^K$ by averaging the absolute sparse codes across all spatial dimensions:

$$u = GAP(Z) = \frac{1}{dhw} \sum_{d'=1}^{d} \sum_{h'=1}^{h} \sum_{w'=1}^{w} \left| Z_{:,d',h',w'} \right|. \tag{7}$$

This vector is then $\ell_1$-normalized for scale invariance, yielding $\tilde{u}$. The attribute proxies $\hat{a} \in \mathbb{R}^{|A|}$ are subsequently obtained by applying the linear transformation from $\tilde{u}$ to establish the relationship between the attribute and the dictionary atoms:

$$\hat{a} = W\tilde{u}. \tag{8}$$

Each row of $W$ selects a small set of atoms that collectively predict one attribute, while $\hat{a}$ serves as both a training signal for attribute regularization (in Section 3.2.1) and a control handle for attribute-conditioned generation (in Appendix A.5).

### 3.2.1 ATTRIBUTE REGULARIZATION

The objective of Attribute Regularization (AR) is to ensure monotonic consistency between the latent proxy value $\hat{a}$ and the ground-truth attribute value $a$ of the samples. Following Cetin et al. (2022), for each attribute $m$, the distance is computed over all sample pairs $(p, q)$ within a batch as:

$$\text{Dist}_{a_m}[p, q] = a_{p,m} - a_{q,m}, \qquad \text{Dist}_{\hat{a}_m}[p, q] = \hat{a}_{p,m} - \hat{a}_{q,m}. \tag{9}$$

The core loss compares the sign structure of these distance matrices:

$$\mathcal{L}_{\text{AR}}^m(W) = \text{MAE}\left(\tanh\left(\delta\text{Dist}_{\hat{a}_m}\right) - \text{sgn}\left(\text{Dist}_{a_m}\right)\right), \tag{10}$$

where $\text{MAE}$ denotes Mean Absolute Error; the $\text{sgn}$ function abstracts the ground-truth differences into a scale-invariant "hard" ordinal target $\{-1, 0, 1\}$; the $\tanh$ function provides a smooth and differentiable "soft" approximation for the predicted differences; $\delta$ is a scaling factor that adjusts the sharpness of this approximation, balancing smooth optimization with faithful ordinal alignment. This deliberate asymmetry allows the model to learn the correct ordering robustly via smooth gradients, without sensitivity to the absolute scale of attributes.

To compute the total attribute regularization loss, we sum the individual losses for all attributes:

$$\mathcal{L}_{AR}(W) = \sum_{m \in A} \mathcal{L}_{AR}^m(W) \tag{11}$$

### 3.2.2 GROUP SPARSITY ON $W$

To encourage a compact and interpretable mapping between attributes and atoms, we apply a group sparsity loss to the mapper matrix $W \in \mathbb{R}^{|A| \times K}$, where rows correspond to attributes and columns correspond to atoms. The sparsity regularization term is defined as:

$$\mathcal{L}_{GS}(W) = \lambda_{\text{row}} \sum_{m=1}^{|A|} |W_{m,:}|_1 + \lambda_{\text{col}} \sum_{n=1}^{K} |W_{:,n}|_1, \tag{12}$$

which drives $W$ toward a two-dimensional block sparse structure, where attributes are explained by distinct, minimally overlapping atom subsets under regularization parameters $\lambda_{\text{row}}$ and $\lambda_{\text{col}}$ for column and row sparsity, respectively.

### 3.3 OPTIMIZATION STRATEGY

To obtain a stable and interpretable mapping between attributes and atoms, we adopt a three-stage optimization strategy. The process first pretrains the model to ensure robust image representation, then imposes attribute-aware sparsity to align atoms with attributes, and finally freezes the learned support to refine the mapping and guarantee interpretability.

**Stage I: Pretraining for Dictionary Stabilization.** In the first stage, we treat the model as a pretrained image representation learner by optimizing only the reconstruction and latent coding losses. This stabilizes the encoder–decoder and dictionary atoms before attribute supervision is introduced:

$$\min \mathcal{L}_{\text{SSC}} \tag{13}$$

**Stage II: Attribute-Regularized Sparse Mapping.** In this stage, we introduce the attribute mapper $W$, designed with row- and column-wise sparsity constraints to enforce one-to-few and consistent associations between attributes and dictionary atoms. To initialize the mapping, $W$ is first estimated

Table 1: Ablation study on the reconstruction accuracy of Attri-SSC-VAE on the EMIDEC dataset, quantified with the maximum mean discrepancy (MMD). The MMD results are given as $\pm$ standard deviation. AR: attribute-regularization.

| Model | VAE | $\beta$-VAE | AR-VAE | Attri-VAE | Attri-SSC-VAE |
|---|---|---|---|---|---|
| **MMD** $\times 10^2 \downarrow$ | $1.86 \pm 0.06$ | $1.38 \pm 0.04$ | $1.74 \pm 0.06$ | $1.18 \pm 0.03$ | $\mathbf{0.64 \pm 0.02}$ |

using linear regression between atom usage vectors and observed attribute values, yielding a least-squares (LS) solution:
$$W_0 = LS(a, W\tilde{u})$$
Starting from this initialization, we optimize $W$ jointly with the SSC-VAE by augmenting the objective with structured sparsity regularization:
$$\min \mathcal{L}_{\text{SSC}} + \mathcal{L}_{GS}(W), \tag{14}$$
where $\mathcal{L}_{\text{GS}}(W)$ imposes row- and column-wise sparsity on $W$, encouraging each attribute to be explained by a small subset of atoms while ensuring that each atom specializes in only a few attributes.

**Stage III: Freezing Row-Support and Refinement.** Once $W$ converges to a sparse structure, denoted by $W_{stage2}$ at the end of Stage II, we freeze the Top-$K$ atoms selected for each attribute, remove the $\ell_1$ penalties, and fine-tune the mapping with a penalty to limit the drift $\Delta W$ to stabilize atom magnitudes. This stage enforces a transparent one-to-few mapping and prevents oscillatory reassignments:
$$\min \mathcal{L}_{\text{SSC}} + \gamma \mathcal{L}_{\text{AR}}(W) + \mu \|\Delta W\|_2^2, \tag{15}$$
where the drift penalty is applied to $\Delta W = W - W_{stage2}$, and $\|.\|_2^2$ denotes the squared Frobenius norm and $\mu$ is the regularization parameter. This staged curriculum produces a model that is both stable and interpretable, with attribute-aligned atoms that support transparent analysis and controlled generative manipulation.

## 4 EXPERIMENTS

The performance of the proposed Attri-SSC-VAE, both qualitatively and quantitatively, was compared with Attri-VAE (Cetin et al., 2022) and its three variants of VAE, $\beta$-VAE, and AR-VAE from the perspective of fine-grained image reconstruction and interpretable medical image generation. This allows us to benchmark our structured sparse coding approach against established continuous latent variable models. For a fair comparison, we use same medical dataset EMIDEC, a collection of medical images well-suited for this task due to its associated clinical attributes, used by Attri-VAE (Cetin et al., 2022). Further details on the dataset specifics and pre-processing steps are provided in Appendix A.2. All fixed hyperparameters used throughout our experiments, including data loading parameters and loss weights, are meticulously detailed in Appendix A.6 in Table 7.

### 4.1 IMAGE RECONSTRUCTION

We evaluate reconstruction performance along three complementary axes: (i) Reconstruction quality. We PSNR and SSIM to measure voxel-level fidelity. (ii)Perceptual quality. We use FID and LPIPS to evaluate perceptual similarity and realism. (iii) Preservation of information and distribution. We quantify preserved information in the latent representation using Mutual Information (MI) and MMD. Full metric descriptions are provided in Appendix A.3.

Table 1 presents a comparative analysis of reconstruction accuracy on the EMIDEC dataset, measured by MMD, where lower values indicate that the distribution of reconstructed images is closer to that of the ground-truth data. Our proposed Attri-SSC-VAE achieves the lowest MMD score of $0.64 \pm 0.02$, substantially outperforming all other models. The closest competitor, Attri-VAE, scores $1.18 \pm 0.03$. This demonstrates that the structured sparse coding backbone effectively preserves fine-grained details essential for high-fidelity reconstruction.

To further evaluate Attri-SSC-VAE, we compare it against the strongest baseline, Attri-VAE, across multiple metrics as shown in Table 2. Our model consistently outperforms the baseline, achieving

Table 2: Performance comparison between Attri-SSC-VAE and Attri-VAE in terms of MI, PSNR, SSIM, FID and LPIPS.

| Model | MI $\times 10^2 \uparrow$ | PSNR $\uparrow$ | SSIM $\uparrow$ | FID $\downarrow$ | LPIPS $\downarrow$ |
|---|---|---|---|---|---|
| Attri-VAE | 1.1310 | 15.1908 | 0.4254 | 145.1451 | 0.2021 |
| Attri-SSC-VAE | **1.3753** | **28.6414** | **0.9371** | **42.1573** | **0.0491** |

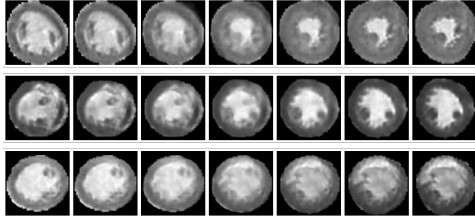
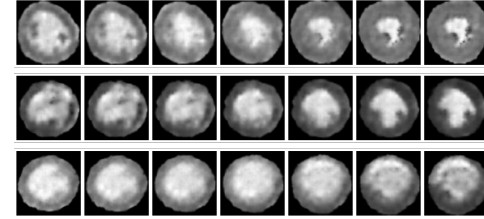

    (a) Interpolation results of our method          (b) Results of the Attri-VAE (Cetin et al., 2022)

Figure 2: Comparison of latent space interpolation results: Left shows results from our method on the EMIDEC dataset, right shows results from the original paper. Each row demonstrates smooth transitions for myocardial thickness (top), scar presence (middle), and cardiac condition (bottom).

a substantial reduction in FID from 145 to 42, reflecting markedly improved perceptual realism. It also attains higher PSNR and SSIM scores, indicating superior pixel-level fidelity, while increased MI confirms better preservation of information in the latent space. The lower LPIPS further demonstrates enhanced perceptual similarity. Together, these results highlight that Attri-SSC-VAE delivers more accurate, informative, and high-quality reconstructions, validating the benefits of combining structured sparse coding with attribute-aware regularization.

### 4.2 Image Generation

For image generation, three sets of experiments are done to assess controllability and semantic alignment of the learned representations: (i) latent-space interpolation, where smooth transitions between two medical images are generated to evaluate continuity and realism; (ii) attribute-guided manipulation, where continuous attributes are gradually varied to examine whether edits are consistent, monotonic, and localized to the relevant anatomical regions; and (iii) attribute correlation analysis, where the learned sets of sparse codes quantitatively capture the attribute corelation of samples.

#### 4.2.1 Latent-space Interpolation

To assess the structural coherence of the learned latent space, we perform linear interpolation between latent representations of distinct test samples. Detailed descriptions of the latent-space interpolation process are provided in Appendix A.4. As illustrated in Figure 2a, Attri-SSC-VAE produces smooth and anatomically plausible transitions, with intermediate images reflecting realistic cardiac states. In contrast, the baseline interpolations in Figure 2b display less coherent semantic changes and blurring images. These results indicate that Attri-SSC-VAE captures a continuous and meaningfully structured latent manifold, which is essential for realistic image synthesis and modeling of disease progression.

#### 4.2.2 Attribute Manipulation in Latent Space

To evaluate fine-grained control over attribute, we systematically vary latent coefficients corresponding to a single clinical attribute while keeping other factors fixed to manipulate the attribute. Details are provided in Appendix A.5. As shown in Figure 3, this targeted manipulation produces precise and clinically plausible edits. For example, increasing the 'LV Volume' coefficient visibly enlarges the left ventricle chamber without affecting unrelated anatomical structures. Corresponding attention maps confirm that changes are correctly localized, demonstrating strong disentanglement. This

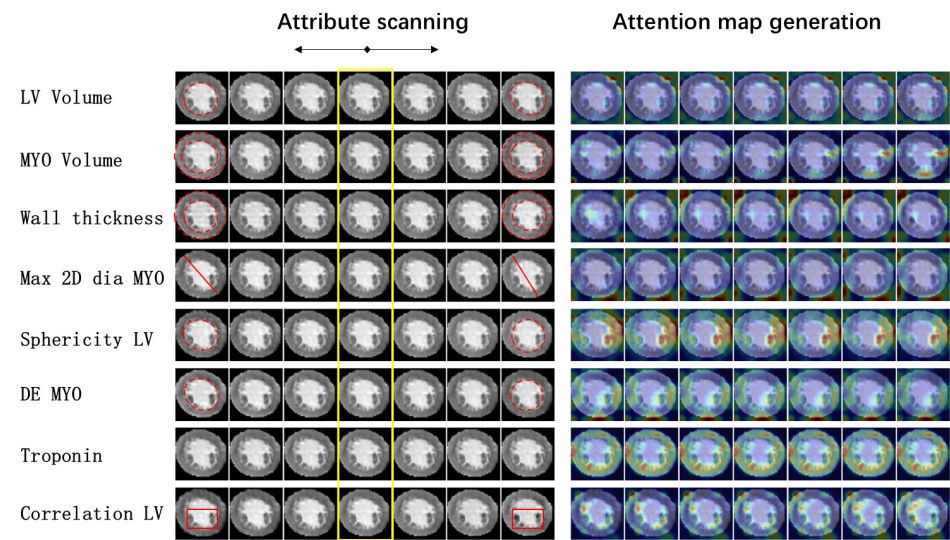

Figure 3: **Controllable generation via bidirectional attribute editing.** Starting from a real image (column in the center, indicated by yellow box), we decrease (left) and increase (right) the latent coefficient for a single attribute (e.g., LV Volume). The resulting edits are precise and localized, demonstrating the model's capacity for highly disentangled and fine-grained control.

Table 3: Interpretable Correlation Analysis of Attribute Pairs.

| Attribute Pair | Ground Truth | | Attri-SSC-VAE | | Attri-VAE | |
|---|---|---|---|---|---|---|
| | $r_{sample}$ | p-value | $r_{recon}$ | p-value | $r_{recon}$ | p-value |
| LV_Volume vs. MYO_Volume | 0.675 | 0.001 | 0.657 | 0.001 | 0.617 | 0.001 |
| MYO_Volume vs. FEVG | -0.271 | 0.147 | -0.146 | 0.442 | -0.515 | 0.004 |

explicit control provides a transparent and interpretable mechanism for simulating hypothetical scenarios and exploring the visual effects of specific pathological features, which is highly valuable in clinical applications.

### 4.2.3 Correlation between Attributes

Our framework captures attribute dependencies by allocating correlated attributes to shared atoms. One positively correlated and one weakly negatively correlated (not significant) attribute pair are presented as examples in Table 3. "Left Ventricle Volume" and "Myocardial Volume" are jointly assigned to atom 313, with a strong ground-truth sample correlation (Pearson $r_{sample} = 0.675$, $p = 0.001$). Using Equation (8) to estimate attributes from the sparse codes, Attri-SSC-VAE preserves this relationship ($r_{recon} = 0.657$, $p = 0.001$), closely matching the ground truth, whereas Attri-VAE slightly underestimates the correlation ($r_{recon} = 0.617$). Conversely, for a weakly negatively correlated pair such as "Myocardial Volume" and "FEVG" ($r_{sample} = -0.271$, $p = 0.147$), Attri-SSC-VAE produces a similarly weak reconstructed correlation ($r_{recon} = -0.146$, $p = 0.442$), while Attri-VAE exaggerates the relationship ($r_{recon} = -0.515$, $p = 0.004$). These results indicate that the one-to-few atom mapping in Attri-SSC-VAE effectively captures clinically relevant attribute correlations, yielding a more faithful and interpretable representation compared to Attri-VAE.

### 4.3 Ablation studies

#### 4.3.1 One-to-few vs. One-to-one Mapping

To examine the effect of one-to-few mapping in Attri-SSC-VAE, we compare it against the one-to-one mapping strategy used in Attri-VAE (Cetin et al., 2022). The results, summarized in Table 4, are evaluated in terms of AR loss and reconstruction quality.

**Attri-SSC-VAE (Full)**: The full model achieves strong overall performance, combining low AR loss with high-fidelity reconstructions, as reflected by PSNR of 28.64 and SSIM of 0.94.

**–Shared Mapping (–W)**: Replacing the shared one-to-few mapping with a one-to-one binding degrades both AR loss and reconstruction performance, showing that shared atoms are better for capturing attribute correlations and enabling attribute alignment.

Table 4: Ablation study results on the contribution of key model components.

| Model Variant | AR-Loss ↓ | PSNR ↑ | SSIM ↑ | FID ↓ | LPIPS ↓ |
|---|---|---|---|---|---|
| **Attri-SSC-VAE** | 1073.18 | 28.64 | 0.94 | 42.16 | 0.049 |
| **w/o shared Mapping (-W)** | 1091.40 | 28.04 | 0.92 | 47.73 | 0.080 |

### 4.3.2 HYPERPARAMETER $\gamma$ SENSITIVITY EVALUATION

Table 5 reports the effect of varying on the trade-off between reconstruction quality and attribute regularization. When $\gamma$ is small (e.g., 0.1 or 1), the model generally maintains good reconstruction performance (PSNR and SSIM remain close to the baseline without AR loss, $\gamma = 0$), but the AR loss remains high, indicating weak alignment between attributes and the latent representation. Introducing AR loss with $\gamma = 5$ achieves the lowest AR loss while preserving reconstruction quality (PSNR and SSIM) and perceptual metrics (FID and LPIPS) compared to the baseline. This demonstrates that adding explicit attribute regularization does not degrade performance while enables attribute-related image editing, generation, and interpretation. However, setting $\gamma$ too high (e.g., 10 or 100) does not further reduce AR loss and instead harms image fidelity, as evidenced by decreased SSIM and degraded perceptual quality. This suggests that overly strong regularization compromises the representational capacity of the latent space.

Table 5: Ablation study results for different values of gamma.

| gamma | AR-Loss ↓ | PSNR ↑ | SSIM ↑ | FID ↓ | LPIPS ↓ |
|---|---|---|---|---|---|
| 0 | - | **29.8703** | **0.9419** | **41.2179** | **0.043** |
| 0.1 | 1371.60 | 28.5174 | 0.9303 | 43.9887 | 0.056 |
| 1 | 1305.92 | 28.0889 | 0.9313 | 45.0348 | 0.061 |
| 5 | **1073.18** | 28.6414 | 0.9371 | 42.1573 | 0.049 |
| 10 | 1285.74 | 27.7333 | 0.9149 | 44.4684 | 0.064 |
| 100 | 1178.67 | 23.8924 | 0.8450 | 55.4306 | 0.154 |

## 5 CONCLUSIONS

In this paper, we introduced Attri-SSC-VAE, a novel framework that combines structured sparse coding with variational autoencoders, guided by explicit clinical attributes. Our method enables interpretable and controllable latent representations, where groups of dictionary atoms capture semantic factors and attribute correlations, allowing precise manipulation of clinically meaningful features. Extensive experiments on the EMIDEC dataset demonstrate that Attri-SSC-VAE achieves superior reconstruction fidelity and generative quality compared to existing baselines, while producing a semantically coherent latent space that supports smooth interpolations and attribute-guided editing. These results highlight the potential of structured, attribute-regularized sparse representations for trustworthy and clinically relevant image generation. While our framework currently relies on labeled attributes and 2D slices, it opens promising directions for semi-supervised attribute discovery, extension to 3D volumetric data, and integration into downstream clinical tasks such as diagnosis or treatment planning.

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

# A APPENDIX

## A.1 NETWORK STRUCTURE

The detailed network structure, including specific layers, resolutions, and channels based on the SSC-VAE backbone, is summarized in Table 6, where it follows a basic structure of encoder-decoder with additional attentive LISTA for structural sparse coding.

## A.2 EMIDEC DATASET

All experiments were conducted on the publicly available EMIDEC dataset(Lalande et al., 2020). EMIDEC dataset is a medical imaging dataset designed for the automatic evaluation of myocardial infarction (MI), with a particular focus on Delayed-Enhancement Cardiac MRI (DE-MRI) sequences. The dataset was collected by the University Hospital of Dijon, France, and aims to advance the research in automated analysis and deep learning methods within the field of cardiac imaging. The dataset includes imaging data from 150 different patients, with 100 cases in the training set, consisting of 33 healthy and 67 pathological cases. The test set includes 50 cases, with 33 pathological and 17 healthy cases. For each case, there is a text file containing clinical information, a NIfTI file with the images, and a NIfTI file with the labeled mask of each area (background, myocardium, cavity, myocardial infarction, and no-reflow).

## A.3 EVALUATION METRICS

We evaluate reconstruction quality along three complementary dimensions: voxel-level fidelity, perceptual realism, and information preservation.

- Voxel-level fidelity:
  - PSNR (Peak Signal-to-Noise Ratio): Measures pixel-wise reconstruction accuracy. Higher values indicate lower reconstruction error and better fidelity.
  - SSIM (Structural Similarity Index): Quantifies structural similarity between reconstructed and ground-truth images, capturing luminance, contrast, and texture consistency.
- Perceptual quality:

Table 6: SSC-VAE backbone.

| Block | Layers | Resolution | Channels |
|---|---|---|---|
| Input | - | $D \times H \times W$ | 1 |
| Encoder | Conv | $D \times H \times W$ | 128 |
| | ResidualBlock(2×Conv + ReLU) | $D \times H \times W$ | 128 |
| | DownSampleBlock(Conv + BatchNorm + MaxPool) | $D/2 \times H/2 \times W/2$ | 512 |
| | ResidualBlock(2×Conv + ReLU) | $D/2 \times H/2 \times W/2$ | 512 |
| | DownSampleBlock(Conv + BatchNorm + MaxPool) | $D/4 \times H/4 \times W/4$ | 512 |
| | ResidualBlock(2×Conv + ReLU) | $D/4 \times H/4 \times W/4$ | 512 |
| | DownSampleBlock(Conv3 + BatchNorm + MaxPool) | $D/8 \times H/8 \times W/8$ | 512 |
| | ResidualBlock(2×Conv + ReLU) | $D/8 \times H/8 \times W/8$ | 512 |
| | NonLocalBlock(4×Conv + Softmax) | $D/8 \times H/8 \times W/8$ | 512 |
| | ResidualBlock(2×Conv + ReLU) | $D/8 \times H/8 \times W/8$ | 512 |
| | GroupNorm + Swish + Conv | $D/8 \times H/8 \times W/8$ | 256 |
| AttentiveLISTA | Conv | $D/8 \times H/8 \times W/8$ | 512 |
| | ResidualBlock(2×Conv + ReLU) | $D/8 \times H/8 \times W/8$ | 512 |
| | ResidualBlock(2×Conv + ReLU) | $D/8 \times H/8 \times W/8$ | 512 |
| | CBAM(ChannelAttention + SpatialAttention) | $D/8 \times H/8 \times W/8$ | 512 |
| Decoder | Conv | $D/8 \times H/8 \times W/8$ | 512 |
| | ResidualBlock(2×Conv + ReLU) | $D/8 \times H/8 \times W/8$ | 512 |
| | NonLocalBlock(4×Conv + Softmax) | $D/8 \times H/8 \times W/8$ | 512 |
| | ResidualBlock(2×Conv + ReLU) | $D/8 \times H/8 \times W/8$ | 512 |
| | ResidualBlock(2×Conv + ReLU) | $D/8 \times H/8 \times W/8$ | 512 |
| | UpSampleBlock(ConvTranspose + Conv) | $D/4 \times H/4 \times W/4$ | 512 |
| | ResidualBlock(2×Conv + ReLU) | $D/4 \times H/4 \times W/4$ | 128 |
| | UpSampleBlock(onvTranspose + Conv) | $D/2 \times H/2 \times W/2$ | 128 |
| | ResidualBlock(2×Conv + ReLU) | $D/2 \times H/2 \times W/2$ | 128 |
| | UpSampleBlock(ConvTranspose + Conv) | $D \times H \times W$ | 128 |
| | GroupNorm + Swish + Conv | $D \times H \times W$ | 128 |

- – FID (Fréchet Inception Distance): Evaluates the distance between feature distributions of reconstructed and real images, with lower scores reflecting more realistic and natural-looking outputs.
  – LPIPS (Learned Perceptual Image Patch Similarity): Assesses perceptual similarity by comparing deep features of paired images. Lower LPIPS indicates higher perceptual resemblance.
- • Distributional alignment and information retention:
  – MMD (Maximum Mean Discrepancy): Measures the statistical distance between the distributions of reconstructed and ground-truth data, where smaller values indicate better global distribution alignment.
  – MI (Mutual Information): Quantifies how much information about the input image is preserved in its latent representation. Higher MI reflects better encoding of clinically relevant details.

### A.4    LATENT-SPACE INTERPOLATION

To qualitatively assess the smoothness of the learned latent manifold, we performed linear interpolation between latent representations of unseen test samples. A smooth and continuous latent space is a key indicator that the model has learned a meaningful representation of the data distribution, rather than simply memorizing the training set.

We begin by sampling two distinct images, denoted as $X_i$ and $X_j$, from the test set. Their respective latent-space representations, $Z_i$ and $Z_j$, are obtained by passing them through the trained encoder network. We then generate a sequence of intermediate latent vectors, $Z_\alpha$, by linearly interpolating between $Z_i$ and $Z_j$:

$$Z_\alpha = (1 - \alpha)Z_i + \alpha Z_j \tag{16}$$

where the interpolation coefficient $\alpha$ is uniformly varied within the range [0,1]. Each interpolated vector $Z_\alpha$ is then fed into the decoder to synthesize a corresponding image $\hat{X}_\alpha$.

This procedure yields a sequence of images that visualizes a traversal path between the two initial points in the latent space. The desired outcome is a sequence where the generated images exhibit a semantically coherent and gradual transition from $\hat{X}_i$ to $\hat{X}_j$. Smooth, anatomically plausible transitions serve as strong evidence that our model has successfully captured the underlying structure of the data, which is critical for robust image synthesis and downstream tasks such as modeling disease progression (Cetin et al., 2022).

## A.5 ATTRIBUTE-AWARE IMAGE GENERATION

A critical evaluation of the Attri-SSC-VAE framework involves assessing its capacity for fine-grained, controllable synthesis of medical images via attribute manipulation. Unlike models with a monolithic latent vector, our framework is explicitly designed to learn a discrete sparse atomic representation of images. The core hypothesis is that through attribute regularization, the model learns to associate specific clinical attributes with distinct, interpretable groups of dictionary atoms. This experiment tests the hypothesis by showing that directly modulating the activation coefficients of attribute-specific atoms, guided by the learned mapper $W$, produces precise and semantically coherent edits to the corresponding visual features in the generated outputs.

The manipulation procedure consists of three main stages:

- **Identifying attribute–atom associations.** We begin with the learned linear mapper $W \in \mathbb{R}^{|A| \times K}$, which links semantic attributes to the sparse coding dictionary for reconstruction of a targeted attribute $\hat{a}$, we select its most influential atoms by choosing the top-$k$ entries with the largest absolute weights $|W_{a,j}|$. Each weight's sign is also recorded, since it determines whether the atom contributes positively or negatively to the attribute, a key factor for directional control.

- **Characterizing valid activation ranges.** To ensure manipulations remain within the natural distribution of the latent space, we estimate the empirical support of each atom using the full training set. For atom $j$, we record its global activation bounds $[Z_j^{\min}, Z_j^{\max}]$ by scanning all spatial locations and training instances. These data-driven bounds prevent manipulations from drifting into out-of-distribution regions, thereby preserving anatomical plausibility.

- **Modulating activations to edit attributes.** Given an input image $X$, we encode it into sparse codes $Z = \text{Encoder}(X)$. To manipulate attribute $a$, we adjust only the activations of its top-$k$ associated atoms while holding all others fixed. The adjustment is controlled by a parameter $\alpha \in [-1, 1]$, sampled at equal intervals, producing the modified activation:

$$Z_j'(\alpha) = (1 - |\alpha|)Z_j + |\alpha|\mathbf{B}_{j,a}(\alpha),$$

for $j \in top-k$, where the target boundary $\mathbf{B}_{j,a}(\alpha)$ is defined as

$$\mathbf{B}_{j,a}(\alpha) = \begin{cases} Z_j^{\max} & \text{if } \text{sign}(\alpha) = \text{sign}(W_{a,j}), \\ Z_j^{\min} & \text{otherwise.} \end{cases}$$

This ensures that increasing $\alpha$ pushes activations toward the boundary that reinforces the attribute, while decreasing $\alpha$ moves them in the opposite direction.

- **Decoding to Image.** The modified sparse codes $Z'$ are decoded to generate the manipulated image $\hat{X}_\alpha$, yielding controlled, semantically consistent edits aligned with clinical attributes.

To visually validate the spatial locus of our manipulation, we generate an attribute-wise attention map for each synthesized image $\hat{X}_\alpha$. Employing a gradient-based saliency method, we compute the influence of the manipulated atoms on the final output. This effectively visualizes the spatial footprint of our intervention, confirming which anatomical regions are being altered.

## A.6 EXPERIMENTAL HYPERPARAMETER SETTINGS

Table 7: Fixed Hyperparameter in our Experiment Settings.

| Hyperparameter | Value | Category |
|---|---|---|
| batch size | 4 | Training setup |
| learning rate (Stage 1 and 3) | $1 \times 10^{-4}$ | Optimizer (Adam) |
| learning rate (Stage 2) | $4 \times 10^{-4}$ | Optimizer (Adam) |
| $\gamma$ | 5.0 | Loss weights (attribute regularization, Eq. (15)) |
| $\delta$ | 10.0 | Loss weights (scaling, Eq. (10)) |
| $\mu$ | 0.01 | Loss weights (Eq. (15)) |
| $\lambda_{row}$ | 10.0 | Loss weights (Stage 2, Eq. (12)) |
| $\lambda_{col}$ | 10.0 | Loss weights (Stage 2, Eq. (12)) |
| K | 512 | Number of Dictionary atoms |
| top-$k$ | 5 | Attribute mapping |

## A.7 LLM USAGE

Writing is polished with the assistance of an LLM.

