# OpenReview forum: "ATTRI-SSC-VAE: Multi-Attribute Regularized Sparse Coding VAEs for Interpretable Medical Image Representation"
_ICLR.cc/2026/Conference — ICLR 2026 Conference Withdrawn Submission_

### Official Review · Reviewer_HSkm · 2025-10-28

**Soundness:** 2
**Presentation:** 1
**Contribution:** 2
**Rating:** 4
**Confidence:** 3

**Summary:**

This paper presents ATTRI-SSC-VAE, a VAE architecture that learns sparse codes for interpretable attributes in medical images, and shows that fine-grained image generation is achievable by controlling and manipulating the attributes during generative procedure.The proposed method is evaluated extensively on a real-life medical dataset of cardiac MRI, demonstrating consistently improved performance over baselines. However, the method seems to be heavily built on existing works (Attri-VAE and SSC-VAE), and in their method section it is not clear to me regarding their contribution architecturally. Besides, more evaluations can be done on other medical datasets with metrics on downstream tasks as well. Hence at this stage I cannot recommend acceptance.

**Strengths:**

Proposed method outperforms baseline considerably in all 6 metrics, and the generated images are visually much cleaner and real-looking. The controlled generation results demonstrate that the different attributes capture the localized region of interests, visually implying the disentanglement in the latent space.

**Weaknesses:**

1. In their method section, it is unclear to me which parts are from existing works and which parts are their distinct contributions. Without this information, it is difficult to judge the overall originality.
2. Evaluations are done exclusively on one dataset and on image generation metric. It would be interesting to see how the methods perform on broader medical datasets, for example skin lesion, lung CT and xrays (they are what I found in the related paper), and how would the learned attributes perform in downstream tasks such as classification and retrieval.
3. The paper needs a more comprehensive related work. Disentanglement and controlled generation in medical images has been investigated by many previous works (non-exhaustive list shown below), but authors have focused mostly on generic VAE models.

- [**Tang et al. 2021**] A disentangled generative model for disease decomposition in chest X-rays via normal image synthesis, Medical Image Analysis, 2021, https://www.sciencedirect.com/science/article/abs/pii/S1361841520302036
- [**Havaei et al. 2021**] Conditional generation of medical images via disentangled adversarial inference, Medical Image Analysis, 2021, https://arxiv.org/pdf/2012.04764
- [**Thermos et al. 2021**] Controllable cardiac synthesis via disentangled anatomy arithmetic, MICCAI, 2021, https://arxiv.org/abs/2107.01748
- [**Hasan et al. 2024**] Semi-Supervised Contrastive VAE for Disentanglement of Digital Pathology Images, MICCAI, 2024, https://papers.miccai.org/miccai-2024/paper/3843_paper.pdf

**Questions:**

1. Overall I find the method section difficult to read. Please highlight the paper's contribution.
2. There are mismatch between text and figures, for example, in Figure 1, there is _manipulation phase_ that is undefined elsewhere, and there is _attribute regularization_ and _representation construction_ in the main forward pass, but the loss function (Eqn (2)) only contains encoder and decoder, are those parts of decoder? Could the author provide pseudo codes for better understanding of the procedure?
3. In Figure 3, I find it difficult to see the difference on generated images, and they all look visually very similar. Could authors elaborate on how to read the red highlight (circles and diagonal lines)? In addition, are there any quantitative measures to justify the claims? Like how much does _LV volume_ increases?
4. In line 150, what are C and C’ corresponding to?
5. Please define the acronyms (various metrics) when first-time using it.
6. Please see weakness number 2. Has the generated images validated on downstream tasks like classification and retrieval?
7. In general, medical image generation is hard to validate due to hallucinations, and it can generate images that look real but not really compatible with human anatomy, so automatic metrics might not be enough. Are the generated images being evaluated by a physician and deemed authentic? Please note that this question will not be part of my decision, but I find it to be very important.

---

### Official Review · Reviewer_HrHD · 2025-11-01

**Soundness:** 2
**Presentation:** 2
**Contribution:** 2
**Rating:** 4
**Confidence:** 2

**Summary:**

The paper proposes Attri-SSC-VAE, which augments an SSC-VAE backbone with  a linear multi-attribute mapper (W) from atom-usage vectors to clinical attributes, and a pairwise ordinal attribute-regularization loss (tanh vs. sgn on pairwise differences), and row/column group sparsity on W to encourage one-to-few attribute–atom bindings. Experiments on EMIDEC cardiac DE-MRI are reported, with large gains over Attri-VAE on PSNR/SSIM, FID/LPIPS, MI, and MMD; qualitative demos include latent interpolations and single-attribute edits; a small correlation analysis (two pairs) suggests the shared-atom design better preserves clinical attribute dependencies.

**Strengths:**

1. Ties from atom activations \(u\) to attributes via a sparse linear map W are straightforward and auditable; the pairwise ordinal loss is scale-free and aligns well with monotone attribute ordering.
2. Building on SSC-VAE with attention-refined LISTA thresholds and CBAM-style refinements is reasonable for detail-preserving reconstructions in medical images.
3. On EMIDEC, Attri-SSC-VAE shows strong deltas vs. Attri-VAE, and MI increases; MMD also drops markedly.

**Weaknesses:**

1. The novelty feels incremental. The core advances of linear attribute heads with sparsity, ordinal pairwise regularization, and SSC-VAE features are composed from known parts (e.g., Attri-VAE-style losses, sparse/structured coding, LISTA). The paper lacks strong ablations disentangling which of SSC-VAE vs. AR vs. sparsity primarily drives the large gains. Current ablations are thin (one mapping variant and a γ sweep).
2. All results are on EMIDEC only. No cross-center generalization, no OOD or robustness tests, and no 3D volume experiments (despite a 3D encoder/decoder notation). This limits claims about clinical reliability and portability.
3. Apart from Attri-VAE/variants and MMD in a small table, there’s no comparison to discrete models (VQ-VAE/VQ-VAE-2/VQ-GAN) or recent concept/CBM approaches adapted to medical images. This undercuts the “structured discrete latents” advantage claims.
4. The pairwise AR loss is O(B$^2$) in batch size; no runtime or memory profile is reported. With K=512 atoms and top-k=5, the method should discuss training/inference costs, convergence behavior across stages, and stability w.r.t. \(K\).
5. Typos (“Attibute-aware” in contributions), grammar error (Line 313 “We PSNR and SSIM”).

**Questions:**

Please refer to the weaknesses part, especially weakness 1, 2, 3. I would like to read the authors' rebuttal and increase my rating if my concerns are sufficiently addressed.

---

### Official Review · Reviewer_KCmJ · 2025-11-01

**Soundness:** 2
**Presentation:** 3
**Contribution:** 2
**Rating:** 2
**Confidence:** 4

**Summary:**

This paper proposed the Attri-SSC-VAE, a method built upon SSC-VAE, that uses attribute data to learn attribute information from sparse coding coefficients. The authors conducted experiments on a medical dataset, compared with the baselines in image reconstruction and image generation.

**Strengths:**

This paper is easy to follow. This work tries to extend the SSC-VAE approach when attribute information is available.

**Weaknesses:**

The innovation of this paper is marginal. This paper proposes the Attri-SSC-VAE, built upon the previous method, SSC-VAE, that uses attribute data to learn attribute information from sparse coding coefficients. The Attribute Regularization (AR) is adopted from the Attri-VAE paper. The Group Sparsity is not novel. The motivation for introducing Group Sparsity is not convincing.

The experiments are weak.
- In Table 1, the authors should also compare with SSC-VAE.
- In the image generation experiments, the authors compared with Attri-VAE. However, this cannot justify the effectiveness of the proposed attribute extraction part. Comparing with Attri-VAE is not fair because Attri-VAE didn't use sparse coding.
- In Figure 3, the attribute editing results are not noticeable. For example, in Figure 3, left, it's hard to see the difference between the images in the LV Volume row and the MYO volume row.
- The authors should demonstrate the effectiveness of the proposed method on a natural image dataset such as the CelebA dataset.

**Questions:**

See Weakness.

---

### Official Review · Reviewer_sXGz · 2025-11-03

**Soundness:** 3
**Presentation:** 3
**Contribution:** 3
**Rating:** 2
**Confidence:** 5

**Summary:**

This paper introduces Attri-SSC-VAE, a potentially newl generative framework for interpretable medical image representation.
The presented model extends the Structured Sparse Coding VAE (SSC-VAE) backbone, which uses a dictionary of latent "atoms" for sparse image representation. Main contribution stems from an attribute regularization mechanism that links "meaningful (?)" clinical attributes (e.g., "LV Volume") to groups of these dictionary atoms, enabling a more realistic "one-to-few" mapping rather than the restrictive "one-to-one" mapping. Experiments on a cardiac MRI dataset (EMIDEC) demonstrate that Attri-SSC-VAE achieves some good reconstruction and generation quality.

**Strengths:**

New Attribute Mapping:
 medical imaging attributes are better represented by groups of latent atoms rather than a single dimension (compared to Attri-VAE), they are proposing Attri-SSC-VAE. (SSC is new)

Reconstruction results seem unbelievable larger than sota (ssim of .93 vs 0.42)

optimization strategy is nicely engineered (three stage training curriculum)

ablation studies are showing the feasibility of the components.

**Weaknesses:**

There are many major weaknesses of this paper, unfortunately, after reading more carefully
-- Image are in 2D. Not 3D.
-- backbone of Attri-VAE is standard VAE backbone while Attri-SSC-VAE uses very powerful backbone. Comparisons are not fair.
-- SSIM values do not seem realistic, 0.42 vs 0.93 is huge gap while visual evaluations only show some smoothing effect in Attri-VAE. This also shows that SSIM should be computed at the image level but with several ROIs, the cardiac images look like very small cropped at LV level, which is not realistic and not suitable for radiology.
-- caveat: only single dataset. There is no generalization power analysis.
-- lack of ablation on some key hyperparams such as k.

**Questions:**

weaknesses above are self-descriptive and should be considered as questions for clarification.

---

### Note · Authors · 2025-11-12

**Comment:**

We would like to withdraw this submission.
After further internal review, we identified issues that require substantial revision and additional experiments.
We will resubmit after improving the work. Thank you for your understanding.

**Withdrawal Confirmation:**

I have read and agree with the venue's withdrawal policy on behalf of myself and my co-authors.

---

### Note · Authors · 2025-11-12

**Comment:**

We would like to withdraw this submission.
After further internal review, we identified issues that require substantial revision and additional experiments.
We will resubmit after improving the work. Thank you for your understanding.

**Withdrawal Confirmation:**

I have read and agree with the venue's withdrawal policy on behalf of myself and my co-authors.